# Applying Reconstructed Daily Water Storage and Modified Wetness Index to Flood Monitoring: A Case Study in the Yangtze River Basin

**Cuiyu Xiao [1], Yulong Zhong [1],*, Yunlong Wu [1], Hongbing Bai [1], Wanqiu Li [2], Dingcheng Wu [3], Changqing Wang [4] and Baoming Tian [1]**

[1]  School of Geography and Information Engineering, China University of Geosciences (Wuhan), Wuhan 430078, China; cyxiao@cug.edu.cn (C.X.); wuyunlong@cug.edu.cn (Y.W.); baihb@cug.edu.cn (H.B.); tianbaoming@cug.edu.cn (B.T.)

[2]  School of Surveying and Geo-Informatics, Shandong Jianzhu University, Jinan 250101, China; 24106@sdjzu.edu.cn

[3]  North Information Control Research Academy Group Co., Ltd., Nanjing 211111, China; wdc1010@live.com

[4]  State Key Laboratory of Geodesy and Earth's Dynamics, Innovation Academy for Precision Measurement Science and Technology, Chinese Academy of Sciences, Wuhan 430077, China; whiggsdkd@apm.ac.cn

*  Correspondence: zhongyl@cug.edu.cn

**Abstract:** The terrestrial water storage anomaly (TWSA) observed by the Gravity Recovery and Climate Experiment (GRACE) satellite and its successor GRACE Follow-On (GRACE-FO) provides a new means for monitoring floods. However, due to the coarse temporal resolution of GRACE/GRACE-FO, the understanding of flood occurrence mechanisms and the monitoring of short-term floods are limited. This study utilizes a statistical model to reconstruct daily TWS by combining monthly GRACE observations with daily temperature and precipitation data. The reconstructed daily TWSA is utilized to monitor the catastrophic flood event that occurred in the middle and lower reaches of the Yangtze River basin in 2020. Furthermore, the study compares the reconstructed daily TWSA with the vertical displacements of eight Global Navigation Satellite System (GNSS) stations at grid scale. A modified wetness index (MWI) and a normalized daily flood potential index (NDFPI) are introduced and compared with in situ daily streamflow to assess their potential for flood monitoring and early warning. The results show that terrestrial water storage (TWS) in the study area increases from early June, reaching a peak on 19 July, and then receding till September. The reconstructed TWSA better captures the changes in water storage on a daily scale compared to monthly GRACE data. The MWI and NDFPI based on the reconstructed daily TWSA both exceed the 90th percentile 7 days earlier than the in situ streamflow, demonstrating their potential for daily flood monitoring. Collectively, these findings suggest that the reconstructed TWSA can serve as an effective tool for flood monitoring and early warning.

**Keywords:** GRACE; terrestrial water storage anomaly; modified wetness index; flood potential index; the Yangtze River basin

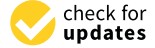



## 1. Introduction

Flooding is a common and severe natural disaster that threatens the lives and property of millions of people worldwide each year [1]. The Yangtze River Basin (YRB) in China, with its abundant water resources, mineral resources, and shipping resources, ensures China's water supply, food, and energy security. Through its governance and development, the YRB supports 459 million people and plays a vital role in economic and social development. However, the YRB in China is a flood-prone area due to climate change and human activities. Flooding has been a significant constraint on socioeconomic development in the YRB. A catastrophic flood event that occurred in the summer of 2020 in the YRB, associated

with the El Niño and Indian Ocean Dipole event [2], affected 38.17 million people and 38,687 km$^2$ of crops, and caused direct economic losses of up to 109.74 billion CNY [3]. Flood monitoring and early warning are crucial for ensuring the safety of people's lives and property and the development of the YRB's economy.

Traditional in situ flood monitoring methods are costly and have limited coverage. Although remote sensing methods can extract inundated areas with high spatiotemporal resolution, they cannot measure variations in terrestrial water storage (TWS) [4], which is fundamental to the regional water cycle [5,6]. The Gravity Recovery and Climate Experiment (GRACE) mission and its successor GRACE-FO can provide information on terrestrial water storage anomaly (TWSA) at a global scale with approximately monthly or higher temporal resolution [7,8], making it a new method for monitoring large-scale flood events. The unique strength of GRACE/GRACE-FO observations for flood monitoring has been demonstrated in the Amazon Basin [9], the YRB [10,11], the Pearl River Basin [12], the Tonlé Sap Basin [13], etc.

Most previous studies have focused on using monthly GRACE data to monitor and predict flood events. However, floods are dynamic hydrological events. High temporal resolution facilitates a detailed understanding of the evolution of flood events, better serving water resources management and solutions. Several recent studies have explored using daily GRACE data to characterize flooding and developed flood indices based on daily GRACE data to indicate flood events. Gouweleeuw et al. [14] first evaluated two daily GRACE gravity field solutions based on a Kalman filter approach using daily river runoff data from the Ganges–Brahmaputra Delta major flood events in 2004 and 2007, demonstrating their potential for sub-monthly flood monitoring. Jäggi et al. [15] defined a wetness index (WI) derived from two daily gravity field solutions and applied this index to the Danube Basin, which showed that WI exceeded the presumed threshold several weeks before the flood peak. Han et al. [16] estimated the daily time series of water storage change in Bangladesh by inverting line-of-sight gravity difference (LGD) between two GRACE-FO spacecraft, which indicated that the daily LGD data can capture rapid water mass change at a sub-monthly time scale. A standardized flood potential index based on ITSG−Grace2018 daily solution was proposed by Xiong et al. [17] and successfully detected 22 sub-monthly exceptional floods and droughts in the YRB between 1961 and 2015. As for the 2020 catastrophic flood in the YRB, Wang et al. [18] combined GRACE data and the high-frequency ground gravity observations of gPhone to characterize 2020 flood events in YRB. Yan et al. [19] used Global Land Data Assimilation System (GLDAS) daily TWS data to investigate the severe Yangtze flooding in 2020. However, the release of ITSG−Grace2018 daily solution has a delay of several months, and the coverage of gPhone is limited. Recently, Xie et al. [20] used machine learning techniques to downscale the GRACE/GRACE-FO TWSA from a monthly scale to a daily scale and further established a normalized daily flood potential index (NDFPI) based on downscaled TWSA. The results showed that NDFPI could robustly and reliably characterize severe flood events at sub-monthly scales.

Humphrey and Gudmundsson [21] reconstructed climate-driven monthly and daily deseasonalized and detrended TWSAs based on historical precipitation and temperature datasets using a statistical model trained with GRACE data. Xiao et al. [22] reconstructed the daily TWSA to monitor the catastrophic flood in northern Henan province of China in July 2021 by introducing the statistical model proposed by Humphrey and Gudmundsson [21]. Result shows that the reconstructed daily TWSA successfully reflected the changes in water storage during the flooding on a daily scale, demonstrating the effectiveness of this reconstruction method for near-real-time floods monitoring. With reference to Humphrey and Gudmundsson [21], Liu et al. [23] proposed a new method to reconstruct monthly climate-driven water storage anomalies that incorporates seasonal terms. On the basis of the model proposed by Liu et al. [23], we reconstruct daily TWSA to analyze the catastrophic flood event in the YRB in 2020. In addition, the WI proposed by Jäggi et al. [15] is calculated on the basis of deseasonalized and detrended TWSA. According to the China Ministry of

Water Resources, the middle and lower reaches of the YRB experiences varying degrees of flood events almost every summer, indicating that main flood event signals mainly exist in the seasonal signal [24]. Therefore, in this study, we retain the seasonal term to calculate the WI, which is defined as the modified wetness index (MWI). Furthermore, we evaluate the reliability of MWI and NDFPI derived from the reconstructed daily TWSA.

The primary objectives of this study are to monitor the 2020 catastrophic flood in the YRB on the basis of reconstructed daily TWSA and evaluate the effectiveness of MWI and NDFPI derived from the reconstructed daily TWSA. The remainder of this paper is divided into five sections. Section 2 describes the study area. The data and methods used in this study are introduced in Section 3. In Section 4.1, we use the TWSA derived from CSR Mascon, ITSG−Grace2018 daily TWSA, soil moisture estimates from the China Land Data Assimilation System (CLDAS-V2.0), and in situ streamflow and vertical displacement from Global Navigation Satellite System (GNSS) stations to assess the reliability of the reconstruction results. Section 4.2 focuses on the spatiotemporal pattern of terrestrial water storage during floods. Furthermore, the capacity of MWI and NDFPI to provide early warning of floods is evaluated in Section 4.3. In Section 5, we compare three flood indices and discuss the limitations of the reconstruction method and possible future applications. Section 6 presents the conclusions of this study.

## 2. Study Region

The Yangtze River is the third largest river in the world, stretching over 6300 km and covering a basin area of approximately 1.8 million km². It originates in the eastern part of the Tibetan Plateau and flows from west to east through 11 provinces before emptying into the East China Sea. The topography of the YRB exhibits a three-step ladder distribution, with high elevation in the west gradually decreasing toward the east [25,26]. Due to its vast territory and complex topography, the YRB has a typical monsoon climate with uneven spatial and temporal distribution of annual precipitation.

Approximately 459 million people reside in the YRB, accounting for one-third of China's population. As a strategic water source for water allocation in China, the YRB is rich in water resources and plays a significant role in China's economic and social development. However, severe flood events have caused devastating impacts on the socioeconomic development of the YRB. In 2020, from June to August, the rainfall in the middle and lower reaches of the YRB was 56% higher than in the same period in the normal year, resulting from extreme weather conditions [27]. The extensive and continuous heavy rainfall caused severe flooding in the middle and lower reaches of the Yangtze River. Combining rainfall and the extent of flooding in the China Flood and Drought Disaster Prevention Bulletin, our study area was delineated (Figure 1). It encompassed a longitude range of 108–122°E and a latitude range of 26–34°N, covering an area of ~360,000 km².

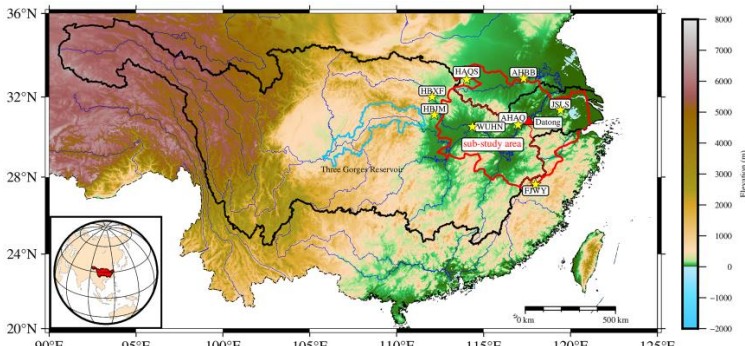

**Figure 1.** Map of the study region. The red line represents the study area boundary defined in this study. The red triangle represents the Datong hydrometric station, and the yellow pentagrams indicate GNSS stations. The sky-blue and black lines represent the Three Gorges Reservoir region and the YRB, respectively. The left part of the study area is the sub-study area.

## 3. Materials and Methods

### 3.1. GRACE Data

The CSR RL06 GRACE/GRACE-FO mascon solutions (CSRM) are provided by the Center for Space Research (CSR) at The University of Texas at Austin and estimated using GRACE Level-1 observations [28]. It is available on the official website as a 0.25° latitude–longitude grid product, while its original spatial resolution is 1° × 1°. The band-limited nature limits the resolution of GRACE/GRACE-FO solutions. It is essential to note that the solutions are limited in resolution, and users should exercise caution when applying them to basins smaller than 200,000 km$^2$.

Recent research has indicated that CSRM data are more consistent with in situ GPS measurements and actual surface mass transport in the YRB than spherical harmonics data and JPL Mascon [29]. As a result, we calibrate our parameters using 163 months of TWSA derived from GRACE CSRM from April 2002 to June 2017.

The ITSG−Grace2018 is a series of GRACE-only gravity field solutions, which are processed based on Level-1B Release 03 data of the GRACE observations [30]. It has an excellent signal-to-noise ratio. We quantitatively evaluated the reconstructed daily TWSA using ITSG−Grace2018 daily TWSA.

### 3.2. Precipitation, Temperature, and Soil Moisture Data

#### 3.2.1. CLDAS-V2.0

CLDAS-V2.0 is a high-resolution (0.0625° × 0.0625°) grid fusion analysis product covering the Asian region (0°–65° N, 60°–160° E) published from 19 January 2017 by the China Meteorological Administration (CMA). This dataset is developed by combining in situ station and satellite observations, employing techniques such as optimal interpolation (OI), probability density function matching, and terrain correction [31]. CLDAS-V2.0 has better quality than its international counterpart in China and a higher spatial resolution. Furthermore, the daily updates of the product with a delay of only 2 days make it possible to reconstruct the daily TWSA in near-real time.

CLDAS-V2.0 includes five products such as atmospheric driving products, surface temperature analysis products, and soil moisture products (divided vertically into five layers: 0–5, 0–10, 10–40, 40–100, and 100–200 cm). In this study, daily precipitation data and 2 m temperature of CLDAS-V2.0 from January 2020 to December 2020 are used to reconstruct TWSA. Soil moisture, one of the main components of terrestrial water storage, is also used to verify our reconstruction.

#### 3.2.2. CGDPA Data

China Gauge-Based Daily Precipitation Analysis (CGDPA) uses about 2400 gauge stations over the Chinese mainland to construct the gridded precipitation using an improved climate-based topography-corrected optimal interpolation method [32]. However, this dataset is only updated to 2019 and is not currently available for download on the official website. Therefore, precipitation data from CLDAS are used instead of CGDPA after 2019.

#### 3.2.3. CN05.1

CN05.1 is a gridded dataset published by the National Climate Centre of the China Meteorological Administration [33]. CN05.1 includes daily average, maximum and minimum temperature, and precipitation. This dataset covers the Chinese mainland with a spatial resolution of 0.25° latitude by 0.25° longitude [34,35] and is constructed on the basis of observations from over 2400 ground-based observation stations in China. It is mainly used for climate change analysis [36]. The daily average temperature data from the CN05.1 dataset are used to reconstruct daily TWSA.

### 3.3. GNSS Observations

In addition to GRACE data and hydrological models, GNSS observations can be used as an independent tool to monitor TWS changes [37,38]. A single GNSS station is sensitive

to load deformation within 100 km [39]. The GNSS time series has high temporal resolution (1 day or shorter) and long-term observations (more than 20 years at some stations), making it a unique advantage for monitoring small-scale surface load deformation.

The precision detrended GNSS time series of eight GNSS stations are obtained from the First Monitoring and Application Center, China Earthquake Administration. This daily product is processed with GAMIT/GLOBK 10.40 software and QACA adjustment. The effects of solid earth tides, ocean tides, and polar tides are removed in the data processing, and the vertical displacement load deformation mainly reflects nontidal atmospheric, oceanic and hydrological influences [40,41]. For comparison with the time series of reconstructed daily TWSA, the nontidal atmospheric loading and nontidal oceanic loading effects are removed from the GNSS time series using the products available from the Earth system modeling group at Deutsches GeoForschungsZentrum.

This study collected time series of eight GNSS stations in and around the study area. Their location distribution is shown in Figure 1. Due to the sparse distribution of the stations, we use the time series of reconstructed daily TWSA from the corresponding grid to compare with the vertical component at a single GNSS station for evaluating our results.

### 3.4. Streamflow Data

The in situ daily streamflow data of Datong station were collected from the Information Center, Ministry of Water Resources, China. These data are used to identify flood events and serve to evaluate the performance of the flood indices. They cover the period from 1 May 2003 to 31 December 2020.

The datasets used in this study are summarized in Table 1.

**Table 1.** Summary of the datasets used in this study.

| Dataset | Time Span | Spatial Resolution | Temporal Resolution | Data Source |
|---|---|---|---|---|
| CSRM RL06 | 2002–2020 | 0.25° | Monthly | http://www2.csr.utexas.edu/grace/RL06_mascons.html, accessed on 2 June 2023 |
| ITSG−Grace2018 | 2003–2020 | 1° | Daily | http://ftp.tugraz.at/outgoing/ITSG/GRACE/ITSG-Grace_operational/daily_kalman/, accessed on 2 June 2023 |
| CLDAS-V2.0 | 2019–2020 | 0.0625° | Daily | https://data.cma.cn/, accessed on 2 June 2023 |
| CGDPA | 2000–2019 | 0.25° | Daily | https://data.cma.cn/, accessed on 2 June 2023 |
| CN05.1 | 2000–2019 | 0.25° | Daily | Contact with the authors |
| GNSS observations | 2020 | stations | Daily | https://www.eqdsc.com, accessed on 2 June 2023 |
| Streamflow | 2003–2020 | stations | Daily | http://xxfb.mwr.cn/sq_djdh.html, accessed on 2 June 2023 |

### 3.5. Methods

#### 3.5.1. Daily TWSA Reconstruction

Humphrey and Gudmundsson [21] trained a statistical model to reconstruct the TWSA using GRACE observations, precipitation, and temperature information. They assumed a water store model where the water outputs are proportional to the water storage and to the residence time of the water storage. Referring to this model, Liu et al. [23] proposed a modified version, which can be expressed as follows:

$$H_{t_i} + d = \left( H_{t_{i-1}} + d \right) \cdot e^{\tau_{t_i}} + P_{t_i}, \tag{1}$$

where $t_i$, $H_{ti}$, and $P_{ti}$ denote the time vector, the reconstructed climate-driven water storage, and the input precipitation during the $t_i$ month, respectively, $e^{\tau ti}$ represents the decay factor

of water storage in the $t_i$ month, and $d$ is the active storage involved in the water cycle over short periods; $\tau$ can be calculated from the following equation:

$$\tau_{t_i} = a + b \cdot P_{t_i}' + c \cdot T_{t_i}', \tag{2}$$

where $a$, $b$, and $c$ are the calibrated model parameters. $P_{ti}'$ and $T_{ti}'$ are the normalized precipitation and temperature, respectively.

Many trends in GRACE are caused by human activities, which the climate-driven model cannot explain by definition [21,42]. Therefore, the linear trend is not included in $H_{t_i}$. On the basis of this model, we reconstruct the daily TWSA at the regional and grid scale.

### 3.5.2. Calculation of MWI

Referring to the formula of Jäggi et al. [15], the formula for calculating *MWI* in this study is as follows:

$$MWI = \frac{TWSA_{\text{detrended}}}{S}, \tag{3}$$

where $TWSA_{\text{detrended}}$ is the reconstructed daily *TWSA*, and $S$ represents the standard deviation of $TWSA_{\text{detrended}}$. For comparison with the NDFPI, we also normalize the *MWI* in this study.

### 3.5.3. Calculation of NDFPI

Reager and Famiglietti [43] proposed a monthly flood potential index (FPI) based on GRACE TWSA and precipitation to indicate flood events. The FPI characterizes the effective water storage capacity of the basin. A daily scale flood potential index (NDFPI) is developed on the basis of FPI [20].

The daily flood potential amount (*DFPA*) can be obtained by subtracting the water storage deficit from the daily precipitation:

$$DFPA(t) = P(t) - (TWSA_{\max} - TWSA(t-1)), \tag{4}$$

where $t$, $P$, $TWSA_{\max}$, and $TWSA(t-1)$ represent the daily time vector, the daily precipitation on the $t$-th day, the maximum of historic storage anomaly time series, and the storage amount from the previous day, respectively.

The NDFPI can be obtained by normalizing the *DFPA*:

$$NDFPI(t) = \frac{DFPA(t) - DFPA_{\min}}{DFPA_{\max} - DFPA_{\min}}, \tag{5}$$

where $DFPA_{\max}$ is the maximum value of *DFPA*, and $DFPA_{\min}$ represents the minimum value of *DFPA* during the study period. *NDFPI* ranges from 0 to 1. A higher value of NDFPI represents a higher risk of flooding in the basin.

The middle and lower reaches of the YRB have a significant long-term trend in TWS over the last two decades due to various factors, such as precipitation and human activities [44]. Several studies have reported an upward trend in TWSA in the YRB [45]. If the TWSA trend increases, the MWI and NDFPI will also increase [24]. Using the TWSA that contains an upward trend term to calculate MWI and NDFPI could result in the omission of prior flood events. As a result, the MWI and NDFPI in this study is calculated with reconstructed TWSA that is detrended.

## 4. Results

### 4.1. Evaluation of the Reconstructed TWSA

The comparison of reconstructed monthly/daily TWSA, monthly TWSA derived from CSRM, and daily TWSA from ITSG−Grace2018 solutions is presented in Figure 2. To facilitate comparison, we remove the trend terms from the CSRM TWSA using a least-squares fitting method [46,47], as we do not reconstruct the trend term. The time series of

reconstructed monthly TWSA and CSRM monthly TWSA show good consistency, with similar phases and amplitudes. The correlation coefficient (CC) between them is 0.90, the Nash–Sutcliffe efficiency coefficient (NSE) is 0.81, and the root-mean-square error (RMSE) is 30.40 mm per month. In addition, there is also a good agreement between the reconstructed daily TWSA and the ITSG−Grace2018 daily TWSA. However, the extremely low values of CSRM TWSA in 2013, 2018, and 2019 are significantly lower than that of reconstructed monthly TWSA.

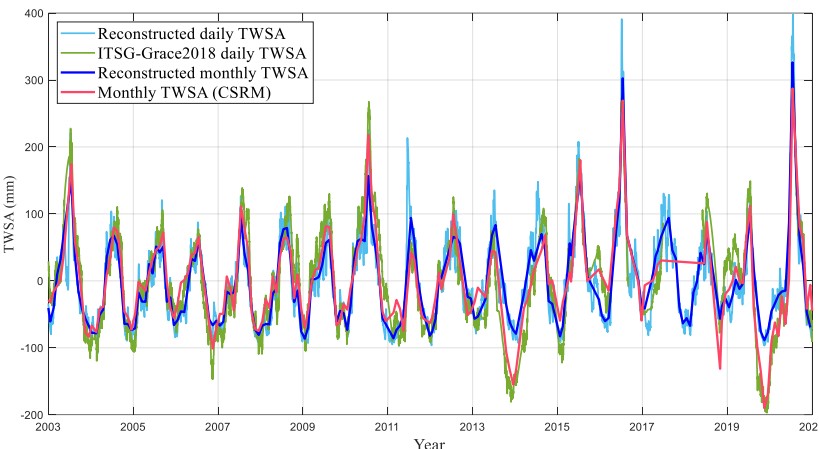

**Figure 2.** Time series of reconstructed monthly/daily TWSA, detrended monthly TWSA from CSRM and detrended daily TWSA from ITSG−Grace2018 solutions in the study area from 2003 to 2020.

To verify the reliability of the reconstruction results, we compare the reconstructed daily TWSAs on the grids where the GNSS stations are located with the time series of vertical displacement of the corresponding GNSS stations and soil moisture. Soil moisture variations show significant positive correlation with the variations in TWS, reaching their peaks and troughs almost simultaneously. However, the time series of soil moisture in all grids shows an abrupt and unusual drop at the beginning of May, which we attribute to the error in the simulation of soil moisture. This phenomenon is not observed in other products, such as the GLDAS and CMA Global Atmospheric Reanalysis (CRA) dataset (Figure S1).

The daily vertical displacements of eight GNSS stations are also compared with reconstructed daily TWSA. The vertical displacement of HBXF station (Figure 3a), WUHN station (Figure 3d), AHBB station (Figure 3e), AHAQ station (Figure 3f), and JSLS station (Figure 3g) agrees relatively well with TWSA, although they differ slightly in the time of reaching the extreme maximal and minimal values. The observations from these five GNSS stations are able to reflect the hydrological load displacement deformation during flooding. The HBJM, HAQS, and WUHN stations also exhibit decreasing in vertical displacements (Figure 3b–d, reversed axis), while there are higher peaks in late August in WUHN (Figure 3d), and early October in HBJM and HAQS (Figure 3b,c).

We further analyze the difference between the time series of GNSS vertical displacements and reconstructed daily TWSA. Firstly, the spatial resolution of the precipitation and temperature data used for the reconstruction is higher than the spatial resolution of GRACE (~300 km), and there may be some improvement in the resolution of the reconstructed TWSA [22]. However, the spatial resolution of the reconstructed TWSA is still coarser than that of GNSS, which results in different spatial sensitivity of GRACE and GNSS to water variations [29,37]. Other geophysical signals contained in the GNSS vertical displacements which are not entirely removed or possible common mode errors may contribute to this phenomenon [40,48]. Additionally, the Three Gorges Reservoir usually begins to store water at the end of September, and the water level reaches 175 m at the end of October each year [49]. HBJM station is located near the Three Gorges reservoir area, and its vertical

displacement is affected by the water impounding in the Three Gorges reservoir. This can partly explain the variation in the time series of vertical displacement at HBJM station.

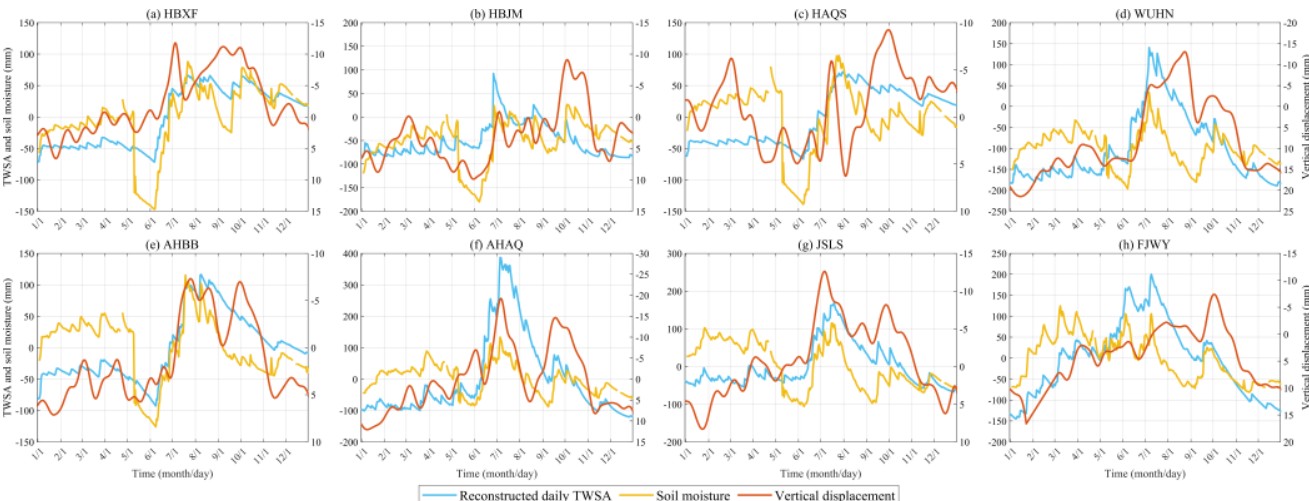

**Figure 3.** Comparison of reconstructed daily TWSA, soil moisture, and the vertical displacement of eight GNSS stations during 2020. It should be noted that the axis of vertical displacements from GNSS are inverted for comparison.

Overall, the GNSS vertical displacements and soil moisture variations provide additional qualitative verification of the reconstruction results.

Since Datong station is located in the center of the study area rather than at the outlet of the basin, the control area of Datong station within the study area is divided as a sub-study area to better compare the reconstructed daily TWSA with the daily streamflow. Figure 4a presents the comparison of the reconstructed daily TWSA of the sub-study area with the in situ daily streamflow from the Datong station during the period 2003 to 2020. Both the regional average TWSA and the streamflow are high in summer and low in winter, exhibiting marked seasonal characteristics, which is consistent with the seasonal variation in precipitation. The dynamics of the reconstructed daily TWSA show good agreement with the changes in streamflow, and the CC between them is as high as 0.87. In particular, the reconstructed daily TWSA is able to capture the significant increase in TWS during the 2016 and 2020 flood events in the YRB.

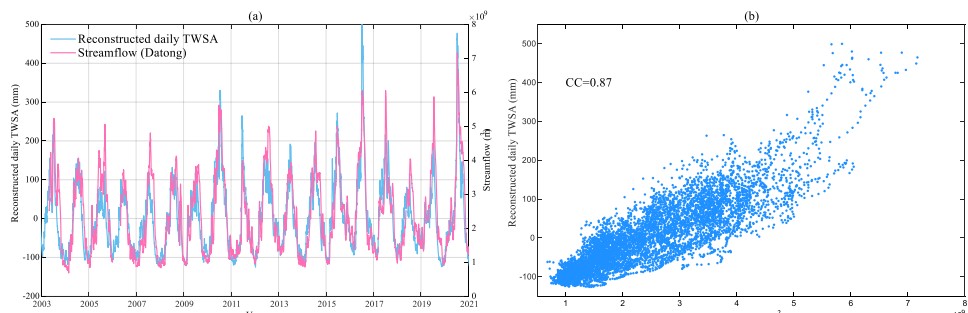

**Figure 4.** (**a**) Time series of reconstructed daily TWSA in the sub−study area and daily streamflow from Datong station from 2003 to 2020; (**b**) scatter graph. The X-axis represents streamflow, and the Y-axis represents TWSA.

### 4.2. Spatial and Temporal Variation in TWSA during the Flood Event

The spatial distribution of reconstructed daily TWSA in the study area from June to August 2020 is displayed in Figure 5. As observed from the figure, most of the grids in the study area exhibit negative TWS anomalies on 1 June, indicating that the region

is relatively dry. With the onset of the plum rains, the TWSAs start to increase rapidly. Some grids in the southern part of the study area begin to show positive TWS anomalies in early to mid-June. By the end of June, almost all the grids in the study area display positive TWS anomalies. Throughout July, the TWS of the grids in the study area continues to show significant positive anomalies, particularly in the middle part. By early August, the range of TWSA exceeding 200 mm in the middle of the study area decreases, and the anomalies finally subside after September. The spatial variations of monthly TWSA from CSRM (Figure 6) are similar to those of reconstructed daily TWSA. However, reconstructed daily TWSA captures the spatial variation in TWS at a daily scale, while monthly TWSA from CSRM reflects the variation at a monthly scale. The finer temporal resolution of the reconstructed TWSA provides more detailed information on the spatial evolution of floods, which in turn facilitates disaster prevention, mitigation, and post-disaster management.

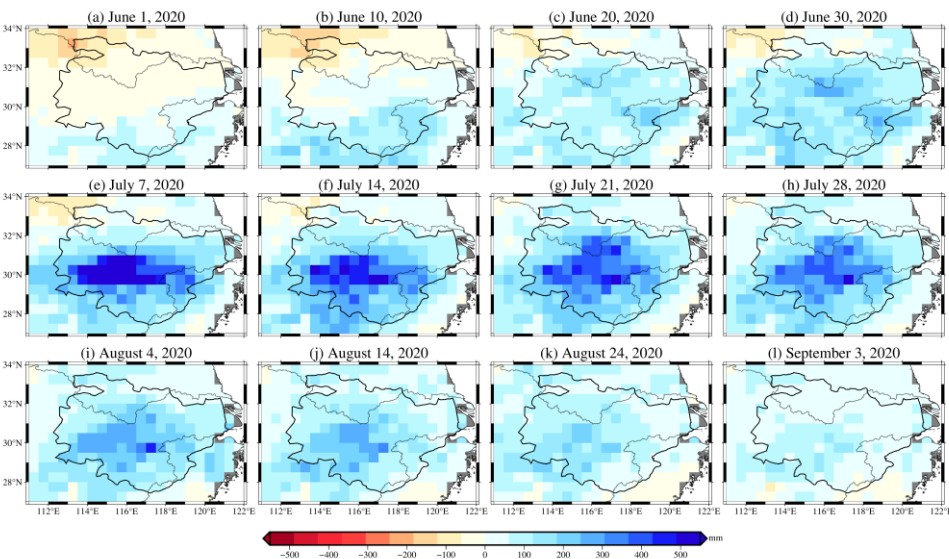

**Figure 5.** Spatial distribution of the reconstructed daily TWSA during the flooding period from 1 June to 3 September 2020.

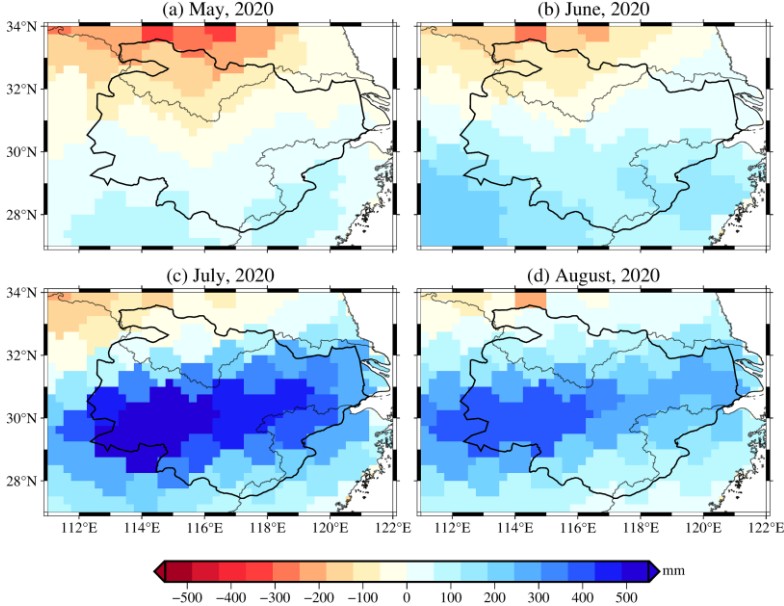

**Figure 6.** Spatial distribution of TWSA derived from CSRM from May to August 2020.

The average TWSA in the study area peaked at 206.34 mm on July 19 (Figure 7). From 1 June to 19 July TWSA increased by 407.57 mm (146.73 km$^3$). After reaching the

peak, the average TWSA of the study area showed a downward trend and decreased to −118.67 mm on 31 August (Figure 7). Although the variations in soil moisture are strongly correlated with the variations in reconstructed TWSA, their magnitudes differ considerably. Such a significant difference between soil moisture and reconstructed daily TWSA during the peak may be attributed to the limited soil water storage capacity. At this time, the soil water storage in the study area is saturated and can no longer increase. In addition, TWS consists of groundwater, surface water, soil moisture, and snow [50–52]. Variations in soil moisture storage account for only a tiny proportion of the variations in TWS in many areas. The variations in groundwater and surface water storage are usually much greater than variations in soil moisture storage [53]. Moreover, the uncertainty of soil moisture simulated by CLDAS, and the limitations of the statistical reconstruction method are among the reasons. Therefore, the difference between TWSA and soil moisture is considered reasonable. From 1 June to early July, the TWSA is consistent with accumulated precipitation. This is probably because most of the accumulated precipitation in the earlier period serves to increase water storage, but the storage capacity of a region is limited, and water storage does not increase beyond a certain point.

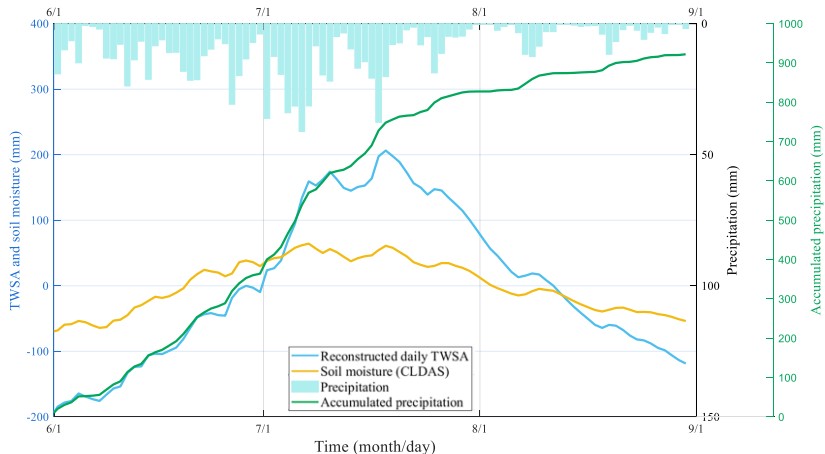

**Figure 7.** Time series of reconstructed daily TWSA, soil moisture, daily precipitation, and accumulated precipitation of the study area during 1 June to 31 August 2020.

### 4.3. Monitoring the Flood Event Using MWI and NDFPI

The storage capacity of a region is limited, and continuous heavy precipitation can saturate regional aquifers. At this point, additional precipitation will increase streamflow if it continues [43]. Although high discharge does not necessarily lead to flooding, it indicates a high risk of flooding. Many studies have used discharge to identify flood levels [20,24,54]. In this study, we use the in situ daily streamflow from Datong station for the 90th percentile flood to evaluate the early warning capability of MWI and NDFPI for the flood event in the YRB in 2020.

The NDFPI is calculated using the reconstructed daily TWSA and daily precipitation data derived from CLDAS V2.0, while the MWI is calculated solely on the basis of reconstructed daily TWSA. The time series of MWI, NDFPI, and daily streamflow from the Datong station in 2020 are presented in Figure 8. The thresholds of the MWI, NDFPI, and daily streamflow for 90th percentile floods are 0.39, 0.49, and $3.95 \times 10^9$ m$^3$, respectively. The results demonstrate that MWI, NDFPI, and streamflow exhibit obvious seasonal characteristics, with higher values during summer and lower values during winter. Both MWI and NDFPI show similar patterns, gradually rising in May, experiencing a rapid increase in June, and peaking in July. MWI and NDFPI indicate that the study area is at higher risk of flooding from June to August, which is consistent with the observed flood evolution recorded in the China Flood and Drought Disaster Prevention Bulletin. MWI and NDFPI exceed their 90th percentile on 22 June, while streamflow exceeds its 90th percentile on 29 June. These two flood indices exceed their thresholds 7 days earlier than streamflow,

indicating that the MWI and NDFPI, based on the daily reconstructed TWSA, effectively provide early warning of this flood event. Figure 8 also illustrates the dynamics of WI for the 2020 flood event. However, WI exceeds its 90th percentile on 1 July, 2 days after the streamflow exceeds its 90th percentile, failing to warn about this flood.

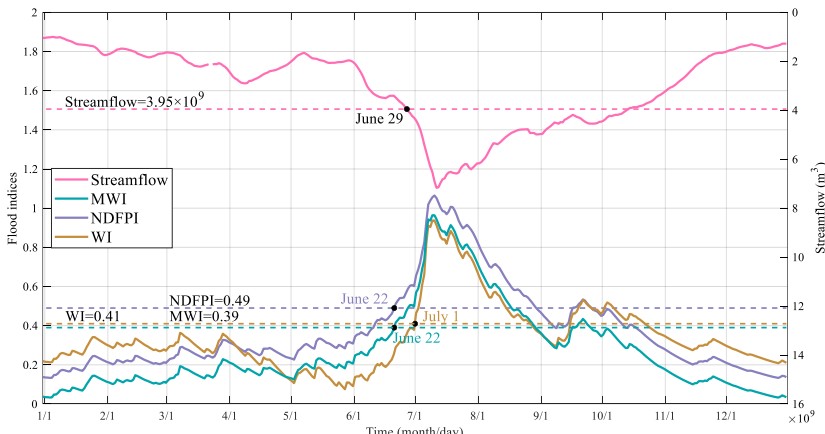

**Figure 8.** Comparison of MWI, NDFPI, and WI (NDFPI is moved up by 0.1 for a better view) of the sub-study area and daily streamflow at Datong station in 2020. The dashed line in the figure represents the thresholds of the MWI, NDFPI, WI, and daily streamflow for the 90th percentile floods from 2003 to 2020.

## 5. Discussion

### 5.1. Comparison of MWI, NDFPI, and WI

TWS can reflect the basin's wet condition, which is closely related to flood generation [43]. WI/MWI is essentially a standardization of the TWSA and reflects the wetness of the basin. A wetter basin leads to greater risk of flooding in the basin. The DFPA quantifies flood risk as the difference between the precipitation input and the water storage capacity on a given day. The difference between daily TWSA and DFPA is $TWSA_{max}$ minus the runoff and evapotranspiration leaving the basin during that day (Equations (4) and (S2)) [55]. While runoff and evapotranspiration may increase during flood events, their numerical values are much smaller than $TWSA_{\max}$. After normalization, the difference between MWI and NDFPI is extremely small, and the two time series almost overlap. Therefore, we shifted the NDFPI up by 0.1. WI is calculated by detrended and deseasonalized reconstructed daily TWSA, and MWI is calculated by detrended reconstructed daily TWSA. Due to the seasonal term, WI is higher than MWI in the spring, autumn, and winter, but lower than MWI in the summer, rising more slowly. Therefore, the WI based on the deseasonalized and detrended TWSA responds more slowly to summer floods than MWI and NDFPI, and it does not even warn about the flood event occurring in the middle and lower reaches of the YRB in the summer of 2020.

Moreover, MWI has an advantage in regions where in situ observations are sparse and precipitation products are less accurate because this index can be directly derived from TWSA observations. The accuracy of the precipitation product also affects the accuracy of our reconstruction results. Therefore, MWI can be calculated with the daily GRACE gravity field solutions that do not depend on precipitation products in such regions.

In addition, we evaluate MWI, NDFPI, and WI for the 90th percentile floods in the 2016 YRB flooding event and the 95th percentile floods in the 2016 and 2020 YRB flooding events (Figure 9). The results show that streamflow exceeds its 90th percentile on 11 May, which is spring in China. In 2016, both MWI and NDFPI exceed their 90th percentile 3 days earlier than streamflow, while WI exceeds its threshold on 19 April, which is earlier than the other two flood indices (Figure 9a). Although the streamflow exceeds the 90th percentile on 11 May, the actual flooding occurs in early July. This might be due to a significant increase in precipitation in the sub-study area from the end of April (Figure S2), which increases the

streamflow and TWS, putting the basin at higher risk of flooding. We also analyzed the 95th percentile floods in 2020 (Figure 9b) and 2016 (Figure 9c). Both MWI and NDFPI exceed the 95th percentile 5 days (20 days) earlier than streamflow in 2020 (2016), demonstrating their effectiveness in flood early warning once again.

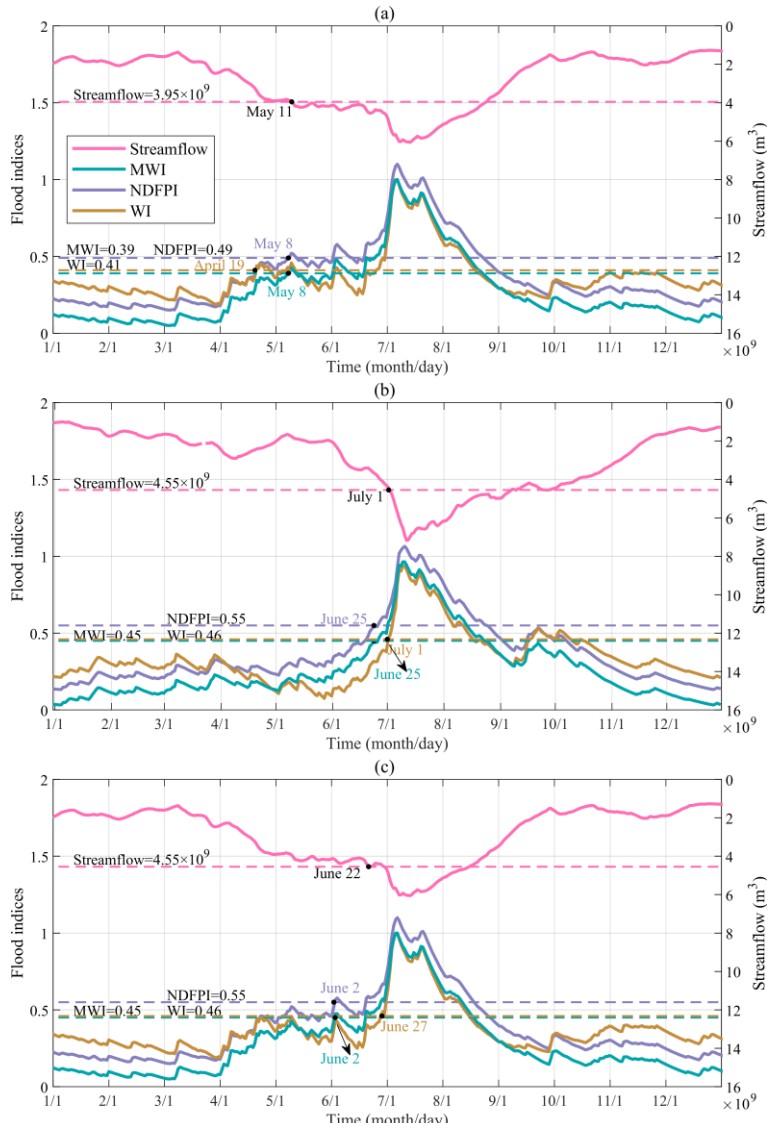

**Figure 9.** Comparison of NDFPI, WI, and MWI (NDFPI moves up by 0.1 for a better view) of the sub-study area and daily streamflow at Datong station in (**a**) 2016 for the 90th percentile floods, (**b**) 2020 for the 95th percentile floods, and (**c**) 2016 for the 95th percentile floods.

### 5.2. Possible Limitations of This Method

The maximum values of the reconstructed monthly TWSA in both 2016 and 2020 are larger than the values of the GRACE TWSA in the same month (Figure 2). Additionally, the low ebb of the reconstructed monthly TWSA in 2013, 2018, and 2019 are also significantly higher than the GRACE TWSA in the same month. According to the China Flood and Drought Disaster Prevention Bulletin, severe flooding or drought events indeed occurred in the middle and lower reaches of the YRB in these years. However, our reconstruction overestimates the TWSA during extreme hydrological events. This is probably because the hydrological processes during extreme hydrological events are different from those during normal periods, and the parameters calibrated using the long-term precipitation, temperature, and monthly GRACE TWSA data may overestimate the residence time of

water in the basin during the extreme events. Therefore, this factor would lead to the overestimation of the reconstructed TWSA during the extreme events.

Furthermore, the deviation of the reconstructed monthly TWSA during drought events shows more obvious deviation than the peak of reconstructed monthly TWSA during flood events compared to the GRACE TWSA (Figure 2). In addition, both our reconstruction and the GRACE observation fail to capture the severe drought events in 2011. Many studies suggested that removing the seasonal term can better identify drought events [56,57]. Many drought indices have also been built on the basis of TWSA with the seasonal term removed [56–58]. If the seasonal term is not eliminated, there will be a wet summer and dry winter, which is not conducive to the identification of drought events. Therefore, the detrended and deseasonalized TWSA can better characterize the drought events in 2011, 2013, 2018, and 2019, although some significant differences still exist at some extreme low values in 2013, 2018, and 2019 (Figure 10).

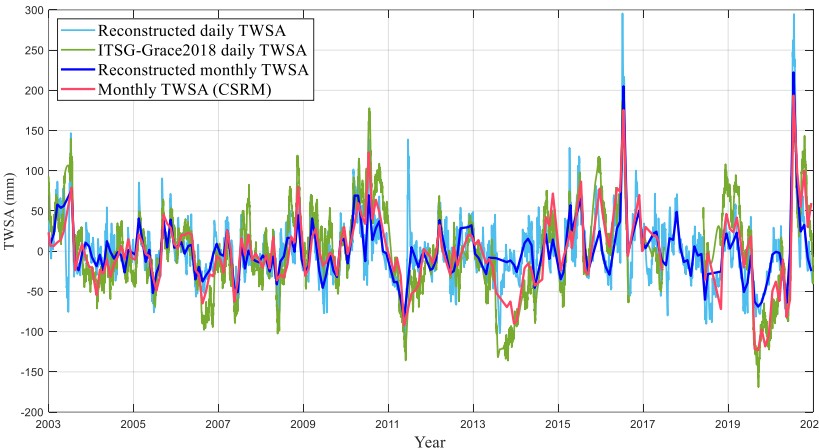

**Figure 10.** Time series of deseasonalized and detrended reconstructed monthly/daily TWSA, monthly TWSA from CSRM, and daily TWSA from ITSG−Grace2018 solutions in the study area from 2003 to 2020.

To improve the reconstruction accuracy, additional variables such as soil moisture can be introduced as input data to calibrate the TWSA reconstruction parameters in the future. Soil moisture affects not only the rate of infiltration and evapotranspiration, but also the generation of subsurface and surface runoff, thus affecting the residence time of water in the basin. When the soil is dry, the infiltration rate is fast, and the evapotranspiration rate is slow. No runoff is generated until the soil moisture reaches the field moisture capacity. When the soil is saturated, all subsequent rainfall, after evapotranspiration loss, generates runoff. Therefore, by incorporating soil moisture to calibrate the decay factor of water storage, more accurate and reliable TWSA reconstruction may be achieved in future studies.

### 5.3. Possible Future Applications

In addition to monitoring flood events, the reconstructed daily TWSA can serve as an input for runoff simulation. Runoff estimation is challenging in areas with harsh environments and limited observations. Basin runoff data are crucial for water resource management and natural disaster monitoring. Previous research utilized a data-driven model to estimate long-term runoff from soil moisture dynamics, with successful results [59]. On the other hand, a study was based on the correlation between in situ river level records and GRACE-based TWS to reconstruct past TWS on an interannual scale [60]. As illustrated in Figure 4b, the reconstructed daily TWSA is highly correlated with in situ runoff observations, which may indicate that reconstructed TWSA has considerable potential for runoff estimation, similarly to soil moisture, with an appropriate model.

## 6. Conclusions

To monitor the changes in TWS during the 2020 flood in the middle and lower reaches of the YRB at a daily scale, we utilized a statistical model to reconstruct the daily TWSA using monthly GRACE observations, daily precipitation, and temperature data. We compared the reconstructed monthly TWSA with monthly TWSA from CSRM, daily TWSA from ITSG−Grace2018 solutions, and in situ daily streamflow at the basin scale. Additionally, the reconstructed daily TWSA at the grid scale was also compared with the time series of vertical displacement from the GNSS stations to assess the reliability of our reconstructed results. Moreover, the temporal and spatial variability of TWS during the 2020 floods in the middle and lower reaches of the YRB was analyzed. Lastly, we compared the in situ streamflow with NDFPI, MWI, and WI calculated on the basis of the reconstructed daily TWSA. From our analyses, the following conclusions can be drawn:

(1) The reconstructed monthly TWSA and CSRM monthly TWSA exhibited overall good consistency, with CC = 0.90, NSE = 0.81, and RMSE = 30.4 mm per month.

(2) The reconstructed daily TWSA in the study area exhibited an increasing trend from early June, reaching a peak of 206.34 mm on 19 July, and then gradually subsiding until September. The reconstructed TWSA successfully captured the daily changes in TWS, while the monthly GRACE data could not capture the dynamic changes in terrestrial water storage within 1 month.

(3) The MWI and NDFPI, calculated from the reconstructed daily TWSA, effectively captured the temporal variation of flooding processes, as both indices exceeded their 90th percentile 7 days earlier than in situ streamflow. This indicates that they are useful for flood monitoring and early warning.

**Supplementary Materials:** The following supporting information can be downloaded at https://www.mdpi.com/article/10.3390/rs15123192/s1: Figure S1. Comparison of three soil moisture products; Figure S2. Accumulated precipitation in 2016 and 2020. Ref. [61] is cited in the Supplementary Materials section.

**Author Contributions:** Conceptualization, C.X. and Y.Z.; data curation, H.B.; formal analysis, C.X., Y.Z. and Y.W.; funding acquisition, Y.Z., Y.W. and C.W.; investigation, C.X.; methodology, C.X. and Y.Z.; validation, C.X. and Y.Z.; visualization, B.T.; writing—original draft, C.X.; writing—review and editing, Y.Z., Y.W., H.B., W.L., D.W., C.W. and B.T. All authors read and agreed to the published version of the manuscript.

**Funding:** This work was supported by the Natural Science Foundation of China (Grant Nos. 42004073, 42174103, and 42274111) and the Open Fund of State Key Laboratory of Remote Sensing Science (Grant No. OFSLRSS202107).

**Data Availability Statement:** The data used in this paper are available as follows: the GRACE/GRACE-FO are published by the Center for Space Research (CSR, http://www2.csr.utexas.edu/grace/RL06_mascons.html, accessed on 2 June 2023), the CLDAS-V2.0 and CGDPA data are provided by the China Meteorological Administration (CMA, https://data.cma.cn/, accessed on 2 June 2023), the GNSS data can be downloaded from the China Earthquake Data Center (CEDC, https://www.eqdsc.com, accessed on 2 June 2023), the geophysical fluid loading products can be downloaded from the Earth System Modeling group at Deutsches GeoForschungsZentrum (ESMGFZ, http://esmdata.gfz-potsdam.de:8080/repository, accessed on 2 June 2023), and the in situ streamflow data were collected from the Information Center, Ministry of Water Resources, China (http://xxfb.mwr.cn/sq_djdh.html, accessed on 2 June 2023).

**Conflicts of Interest:** The authors declare no conflict of interest.

## Abbreviations

The following abbreviations are used in this manuscript:

**Acronyms**

| | |
|---|---|
| TWS | Terrestrial water storage |
| TWSA | Terrestrial water storage anomaly |
| GRACE | Gravity Recovery and Climate Experiment |

| GRACE-FO | GRACE Follow-On |
|---|---|
| GNSS | Global Navigation Satellite System |
| WI | Wetness index |
| MWI | Modified wetness index |
| FPI | Flood potential index |
| NDFPI | Normalized daily flood potential index |
| DFPA | Daily flood potential amount |
| YRB | Yangtze River Basin |
| LGD | Line-of-sight gravity difference |
| GLDAS | Global Land Data Assimilation System |
| CLDAS | China Land Data Assimilation System |
| CSRM | Center for Space Research mascon |
| CMA | China Meteorological Administration |
| CGDPA | China Gauge-Based Daily Precipitation Analysis |
| CC | Correlation coefficient |
| NSE | Nash–Sutcliffe efficiency coefficient |
| RMSE | Root-mean-square error |
| CRA | CMA Global Atmospheric Reanalysis |

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
