# Peer review of "Applying Reconstructed Daily Water Storage and Modified Wetness Index to Flood Monitoring: A Case Study in the Yangtze River Basin"

_remotesensing, doi:10.3390/rs15123192_

Round 1

Reviewer 1 Report

Comments on “Applying reconstructed daily water storage and modified wetness index to flood monitoring: A case study in the Yangtze River Basin (No. 2407476)” by Xiao et. al. This paper employed a statistical model to reconstruct daily terrestrial water storage (TWS) by combining monthly GRACE observations with daily temperature and precipitation data, and the reconstructed daily TWS anomaly was compared with GNSS vertical displacements and utilized to monitor the catastrophic flood event occurred in the middle and lower reaches of the Yangtze River basin in 2020. Furthermore, a modified wetness index (MWI) and a normalized daily flood potential index (NDFPI) were introduced and compared with in-situ daily streamflow to assess their potential for flood monitoring and early warning. I believe that the goal of this study is worth pursuing and meaningful, and the findings can be informative to the readers of Remote Sensing. Therefore, I recommend that this paper could be published after a minor revision.

The comments are provided below, and I hope they can help the authors further improve the manuscript. 

1. For the comparison or validation of the reconstructed daily TWS anomaly, I suggest the authors also can add the results from the daily GRACE solutions (e.g., ITSG-GRACE2018), or the TWSA from hydrometeorological data based on the water budget equation, to conduct a quantitative evaluation.

2. Table 1 is not mentioned in the text, please add a description of Table 1 in the main text.

3. In equation 2, the parameter is not included as the calibrated model parameters.

4. In line 400, there is an error cited for a formula.

5. Lines 413-414, please rephrase “Therefore, daily GRACE gravity field solutions that do 413 not rely on precipitation products to obtain can be an alternative in such regions.

The quality of English is good.

Author Response

My reply is provided in the word file.

Reviewer 2 Report

Manuscript ID: remotesensing-2407476- Applying reconstructed daily water storage and modified wetness index to flood monitoring: A case study in the Yangtze River Basin

 Dear editor

The present manuscript reconstruct daily total water storage (TWSA) to evaluate a flood event in the Yangtze River Basin, China in 2020. By using a combination TWS from GRACE satellite and daily temperature and precipitation data, the authors also compared this data with the vertical displacement with stations at grid scale. Two other indices were also used for comparing their results. The results indicated that estimated TWSA well represented the changes in water storage during the floods showing a good potential for flood monitoring. The manuscript is well written and organized and it has practical application for land management. I have few general comments to be addressed:

- Material and Methods and Result should be written in past tense (e.g. lines 101, 199, 261)

- Explain acronyms when used the first time (e.g Lines 151-184), regarding to this, when it is possible avoid the excess use of acronyms in one sentence, this can be confusing for the reader sometimes

-When mention ‘other studies suggested” please add the corresponding cite (e.g 444)

The manuscript needs minor English editing  

Author Response

My reply is provided in the word file.

Reviewer 3 Report

Dear authors

pls find my comments in the attached pdf

Author Response

My reply is provided in the word file.
